# Liquid phase blending of metal-organic frameworks

Louis Longley[1], Sean M. Collins [1], Chao Zhou [2], Glen J. Smales[3,4], Sarah E. Norman[5], Nick J. Brownbill[6], Christopher W. Ashling[1], Philip A. Chater [4], Robert Tovey [7], Carola-Bibiane Schönlieb[7], Thomas F. Headen[5], Nicholas J. Terrill [4], Yuanzheng Yue[2,8,9], Andrew J. Smith [4], Frédéric Blanc [6,10], David A. Keen [5], Paul A. Midgley[1] & Thomas D. Bennett [1]

The liquid and glass states of metal–organic frameworks (MOFs) have recently become of interest due to the potential for liquid-phase separations and ion transport, alongside the fundamental nature of the latter as a new, fourth category of melt-quenched glass. Here we show that the MOF liquid state can be blended with another MOF component, resulting in a domain structured MOF glass with a single, tailorable glass transition. Intra-domain connectivity and short range order is confirmed by nuclear magnetic resonance spectroscopy and pair distribution function measurements. The interfacial binding between MOF domains in the glass state is evidenced by electron tomography, and the relationship between domain size and $T_g$ investigated. Nanoindentation experiments are also performed to place this new class of MOF materials into context with organic blends and inorganic alloys.

[1] Department of Materials Science and Metallurgy, University of Cambridge, Charles Babbage Road, Cambridge CB3 0FS, UK. [2] Department of Chemistry and Bioscience, Aalborg University, DK-9220 Aalborg, Denmark. [3] Department of Chemistry, University College London, Gordon Street, London WC1H 0AJ, UK. [4] Diamond Light Source Ltd, Diamond House, Harwell Science and Innovation Campus, Didcot OX11 0DE, UK. [5] ISIS Facility, Rutherford Appleton Laboratory, Harwell Campus, Didcot OX11 0QX, UK. [6] Department of Chemistry, University of Liverpool, Crown Street, Liverpool L69 7ZD, UK. [7] Department of Applied Mathematics and Theoretical Physics, Centre for Mathematical Sciences, Wilberforce Road, Cambridge CB3 0WA, UK. [8] State Key Laboratory of Silicate Materials for Architectures, Wuhan University of Technology, 430070 Wuhan, China. [9] School of Materials Science and Engineering, Qilu University of Technology, 250353 Jinan, China. [10] Stephenson Institute for Renewable Energy, University of Liverpool, Crown Street, Liverpool L69 7ZD, UK. Correspondence and requests for materials should be addressed to T.D.B. (email: Tdb35@cam.ac.uk)

Metal–organic frameworks (MOFs), or networked structures of inorganic nodes connected by organic ligands, are flexible materials[1, 2] that can be broadly separated into two classes according to their porosity. Those that contain high internal surface areas are of intense interest for gas separations and catalysis[3–6], while dense MOF materials are investigated for their potential in other applications, e.g., conduction and magnetism[7–9].

The zeolitic imidazolate framework (ZIF) family of MOFs contains structures of tetrahedral $M^{n+}$ nodes, (M = e.g., Zn, Co, Li, B, Ni, Mg) linked through the N atoms of imidazolate ligands[10–12]. Several members, e.g. ZIF-4-Zn, have been observed to possess accessible melting temperatures ($T_m$) between 400 and 600 °C[13]. The melting process proceeds via a dissociation–association mechanism of Zn-N coordination bonding and associated ligand switching between $Zn^{2+}$ centres[14]. This becomes sub-nanosecond at $T_m$, in a manner analogous to the switching between hydrogen bonds in liquid water.

Such liquid states will be of particular intrigue in the development of alternatives to solid-state compounds for industrial-scale gas sorption and separations, due to the better handling and ease of installation compared to their classical solid-state counterparts[15]. Additionally, the intrinsic instabilities of microcrystalline MOF structures often preclude processing into the physical forms and bodies required by industry[16, 17]. Solution casting techniques combine the processability of organic polymers with selective MOF additives[18], though drop-casting, fibre drawing or melt spinning of single-component MOF liquid states would circumvent chemical compatibility concerns.

Cooling of (strongly associated) MOF liquids from above $T_m$ results in a family of melt-quenched glasses chemically different from the inorganic, organic and metallic glass categories known at present. Accordingly, the formation of the liquid and glass phases of MOFs has recently emerged as a new area in an ever-expanding field[13, 14, 19–22]. The reactivity of these 'MOF liquids' has not yet been studied. Possibilities also exist in the production of novel MOF glasses, given the potential to incorporate multiple, designed chemical functionalities within a single glass, or in the creation of hybrid equivalents of alloys, blends and ceramics. Progress in the preparation of crystalline materials containing multiple inorganic or/and organic functionalities within a single framework structure has already been made[23, 24]. These multivariate MOFs[25, 26] arise from the interaction of several chemical components during solvothermal or mechanochemical synthesis, though not in the liquid state.

Here we are interested in how a MOF liquid behaves when combined with a secondary MOF component and the fundamental possibilities that this may afford in new materials' discovery. Specifically, we hypothesize that this may result in the formation of a glass containing interlocking MOF domains. Motivated by the concept of forming this type of material, which we term 'MOF blends', we investigated the high temperature reactions within mixtures of ZIF-4 [M(Im)$_2$] and ZIF-62 [Zn (Im)$_{1.75}$(bIm)$_{0.25}$] (M = Co$^{2+}$, Zn$^{2+}$, Im: C$_3$H$_3$N$_2$$^-$, bIm: C$_7$H$_5$N$_2$$^-$). Previously, it has been observed that, upon heating, both ZIF-4-Zn and ZIF-4-Co undergo a transition to a high-density amorphous phase and a dense crystal on heating to 300 °C and 450 °C, respectively. The zinc framework melts at 550 °C, unlike the dense cobalt crystal, which remains intact until thermal decomposition at ca. 570 °C. ZIF-62 remains in the room temperature crystalline state until liquid formation at 410 °C[13, 27].

## Results

**Differential scanning calorimetry.** Samples of ZIF-4-Zn and ZIF-62 were synthesized and evacuated according to previously

reported solvothermal procedures (Fig. 1a)[27–29]. A physical mixture of the two frameworks in equal weight portions, hereby referred to as (ZIF-4-Zn)(ZIF-62)(50/50), was prepared by ball-milling to ensure sample homogeneity (see Methods). Differential scanning calorimetric (DSC) experiments were then performed up to 590 °C in an argon atmosphere, beyond which thermal decomposition of the liquid state occurred. The first endothermic feature at 225 °C is coincident with a mass loss of ca. 9% and ascribed to desolvation. As expected, two endothermic features belonging to the respective melting points of ZIF-62 and ZIF-4-Zn (445 °C and 580 °C, respectively) were noted, identical to those recorded from pure samples (Fig. 1b)[13]. The melting enthalpy of ZIF-62 was recorded as ca. 3 kJ mol$^{-1}$. Quenching after isothermal treatment for 2 min at 590 °C yielded a glassy, amorphous product (Supplementary Figure 1).

Re-heating of this amorphous sample revealed a single glass transition, glass transition temperature ($T_g$) = 306 °C (Fig. 1b, blue solid line), whereas two separate features at 292 °C (ZIF-4-Zn) and 318 °C (ZIF-62) would have been anticipated[13]. A physical mixture of the two glasses formed separately yielded the expected two $T_g$s (Supplementary Figures 2 and 3). Such a markedly different, single value is indicative of liquid phase mixing, as is also the case in e.g. metallic glasses[30], inorganic oxides and phosphates[31], or miscible polymer blending in organics[32]. We name the blend produced (ZIF-4-Zn)$_{0.5}$(ZIF-62)$_{0.5}$. The ability to tailor $T_g$ was explored through analysis of a further set of (ZIF-4-Zn)$_{1-x}$(ZIF-62)$_x$ mixtures. The results from DSC experiments on the glasses formed upon quenching the liquids from 590 °C (Supplementary Figure 4) show a composition-dependent shift in $T_g$ (Fig. 1c). The increase in $T_g$ with increasing ZIF-62 content follows a linear relation, analogous to the trends observed in binary organic mixtures exhibiting mass additivity behaviour ($\Delta T_g = 0$) e.g. poly(1,3-trimethylene adipate) and poly(vinyl methyl ether)[32].

In order to facilitate the use of electron microscopy as a characterization technique for the blended glass, a physical mixture of ZIF-4-Co and ZIF-62, hereby referred to as (ZIF-4-Co)(ZIF-62)(50/50), was analysed. A pure sample of ZIF-4-Co was synthesized by following prior literature[33]. As expected[27], it possesses a stable amorphous region from 325 to 500 °C (Supplementary Figure 5), before the expected recrystallization to a dense ZIF at ca. 510 °C. No melting above this temperature is observed. DSC experiments on (ZIF-4-Co)(ZIF-62)(50/50) confirmed these transitions, along with the expected $T_m$ of ZIF-62 (Fig. 1d). Quenching of the sample from 425 °C, i.e. a region containing amorphous ZIF-4-Co and liquid ZIF-62, yielded a glass (Supplementary Figure 6). A subsequent DSC of the quenched glass again demonstrated a single $T_g$, at ca. 300 °C (Supplementary Figure 7), despite the fact that it was formed from an interaction between an amorphous solid and a liquid. A second measurement using a slower heating rate again yielded only one $T_g$ (Supplementary Figure 1). Differences between (ZIF-4-Co)$_{0.5}$(ZIF-62)$_{0.5}$ and (ZIF-4-Zn)$_{0.5}$(ZIF-62)$_{0.5}$ are perhaps expected to be small, given the very high viscosities for both ZIF-4-Zn and ZIF-62 reported previously[14, 34].

**Structural characterization.** Small-angle X-ray scattering (SAXS) has previously been used to reveal information on the pore surface and characteristics of MOF-5[35], HKUST-1[36] and monitor particle evolution and growth in situ[37, 38]. Combined with wide-angle X-ray scattering (WAXS), it provides a powerful tool that has also been used to study the collapse of some MOFs to amorphous states[21].

The temperature-resolved WAXS profile of ZIF-62 (Fig. 2a) shows consistent Bragg diffraction from the sample, which

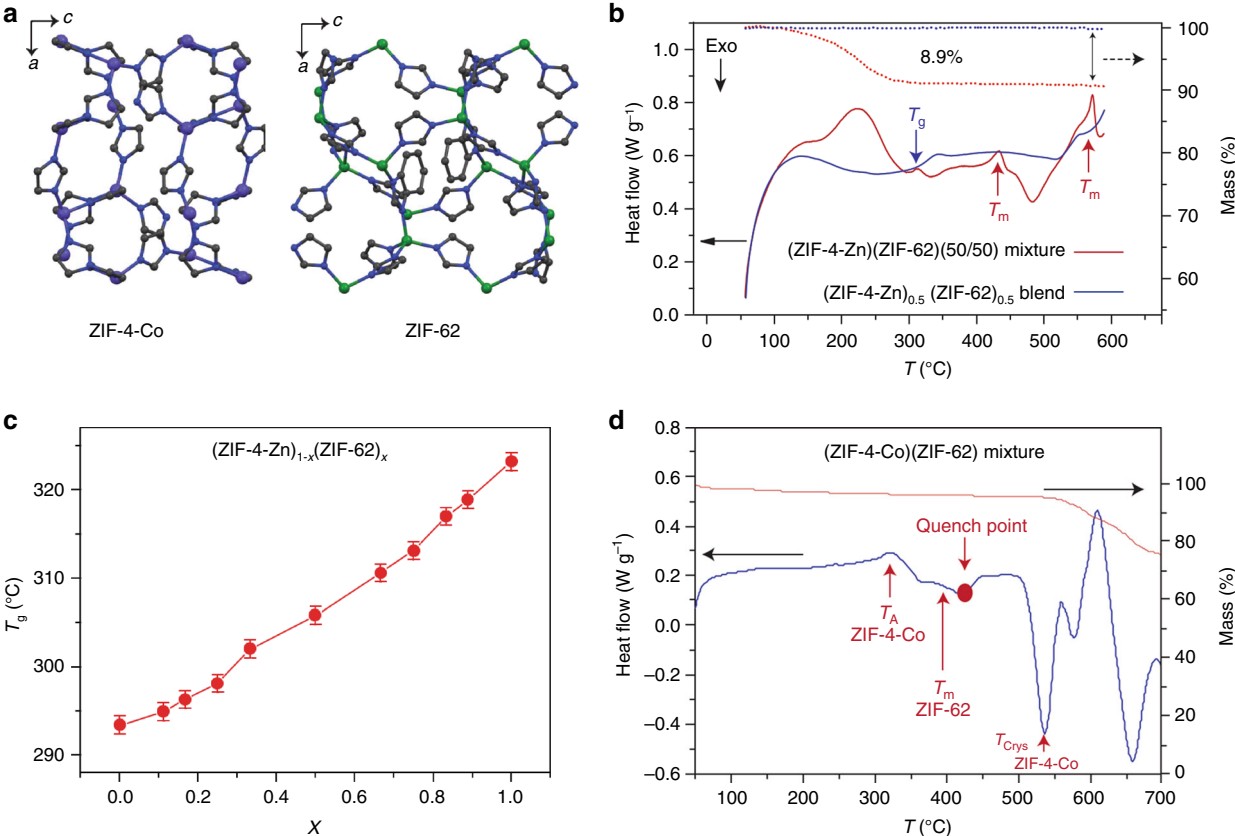

**Fig. 1** MOF liquid dynamics and tailoring glass transition temperature. **a** View down the *b* axis of the unit cells of ZIF-4-Co and ZIF-62. N—dark blue, C—grey, Zn—green, Co—purple, H atoms omitted for clarity. **b** Enthalpy response (red curve) and mass change (dotted curve) in the physical mixture (ZIF-4-Zn)(ZIF-62)(50/50) during heating at 10 °C/min. Blue curve: reheating curve representing the enthalpy response of the corresponding glass that forms upon quenching, i.e., $(ZIF-4-Zn)_{0.5}(ZIF-62)_{0.5}$ during prior cooling at 10 °C min$^{-1}$. **c** Evolving glass transition of the sample series $(ZIF-4-Zn)_{1-x}(ZIF-62)_x$. **d** Enthalpy response (blue curve) and mass change (orange curve) of the physical mixture (ZIF-4-Co)(ZIF-62)(50/50) during heating at 10 °C min$^{-1}$

reduces in intensity and then disappears at ca. 340 °C upon gradual formation of the liquid state. Decomposition of this MOF liquid is then evidenced at ca. 550 °C by the emergence of several Bragg features at relatively large $q$ values. The temperature-resolved WAXS profile of $(ZIF-4-Co)_{0.5}(ZIF-62)_{0.5}$ (Fig. 2b) contains a region in which amorphous ZIF-4-Co and the ZIF-62 liquid are co-existent, between ca. 340 °C and ca. 400 °C. Recrystallization of amorphous ZIF-4-Co to a dense phase is then observed. These observations are broadly consistent with the DSC results presented in Fig. 1, though these differ because of the dissimilar temperature–time profiles of the two experiments.

The decay in SAXS signal at room temperature was extracted from the three-dimensional, variable temperature plot of the SAXS intensity $I_{SAXS}$ for ZIF-62 and follows power law behaviour of the form $q^{-\alpha}$, where $\alpha = 3.9$ (Supplementary Figure 9). At ca. 440 °C, a decrease to $\alpha = 3.4$ is observed, consistent with the formation of rougher internal surfaces upon melting. Computation of the volume-weighted fraction of the particles (Supplementary Figure 9) shows an initial expansion in particle radii from 5 nm at the point of melting, which is consistent with interfacial particle coalescence. The radii then drop drastically and the volume fraction tends to zero, as homogeneous melting of the sample occurs. The increase in particle size at ca. 460 °C then marks the onset of gradual thermal decomposition.

The variable temperature plot of the SAXS intensity $I_{SAXS}$ for (ZIF-4-Co)(ZIF-62)(50/50) (Fig. 2c) was also fitted and displays a lower initial value of $\alpha = 3.66$, consistent with the presence of different internal pore structures and particle sizes within the ball-milled mixture of MOFs. This value increases to 4 on heating

to 340 °C when ZIF-4-Co amorphizes, before decreasing to 3.1 due to both recrystallization of ZIF-4-Co and melting of ZIF-62. The volume-weighted fraction of the particles also reveals that the distribution of particle scatterers is much broader in the initial instance, consistent with the inhomogeneity in sample composition. Like ZIF-62, the particles disappear rapidly upon liquid formation at 340 °C. The broad distribution of particles that starts to appear at ca. 450 °C is ascribed to the known formation of crystallites of a dense ZIF from ZIF-4-Co at these higher temperatures (Fig. 2d).

Liquid-state $^1H$ nuclear magnetic resonance (NMR) was carried out by digesting samples of (ZIF-4-Co)(ZIF-62)(50/50) and $(ZIF-4-Co)_{0.5}(ZIF-62)_{0.5}$ (produced by quenching from 445 °C) in a mixture of deuterium chloride (DCl; 35%)/deuterium oxide (D$_2$O; 100 μL) and DMSO-d$_6$ (500 μL) (Supplementary Figure 10). Resonances in both spectra are fairly broad, arising from the substantial paramagnetic broadening induced by the presence of Co$^{2+}$, predominantly in an octahedral complex coordinated by either H$_2$O or dimethyl sulphoxide (DMSO)[39] giving the metal centre a likely electronic arrangement of $t_{2g}^5 e_g^2$ and three unpaired electrons[40]. This prevents the integration of most of the aromatic signals of the imidazolate ligands. Both NC*H*N$_{Im}$ and NC*H*N$_{bIm}$ peaks are, however, well resolved in the 9–9.7 ppm high-field region and are used to determine the Im: bIm ligand concentration ratios; values of 1:0.076 ± 0.010 and 1:0.054 ± 0.015 are obtained for (ZIF-4-Co)(ZIF-62)(50/50) and $(ZIF-4-Co)_{0.5}(ZIF-62)_{0.5}$, respectively (Supplementary Figure 10). Within error, these values are both in agreement with the expected 1:0.066 stoichiometric ratio. Additionally, in the glass

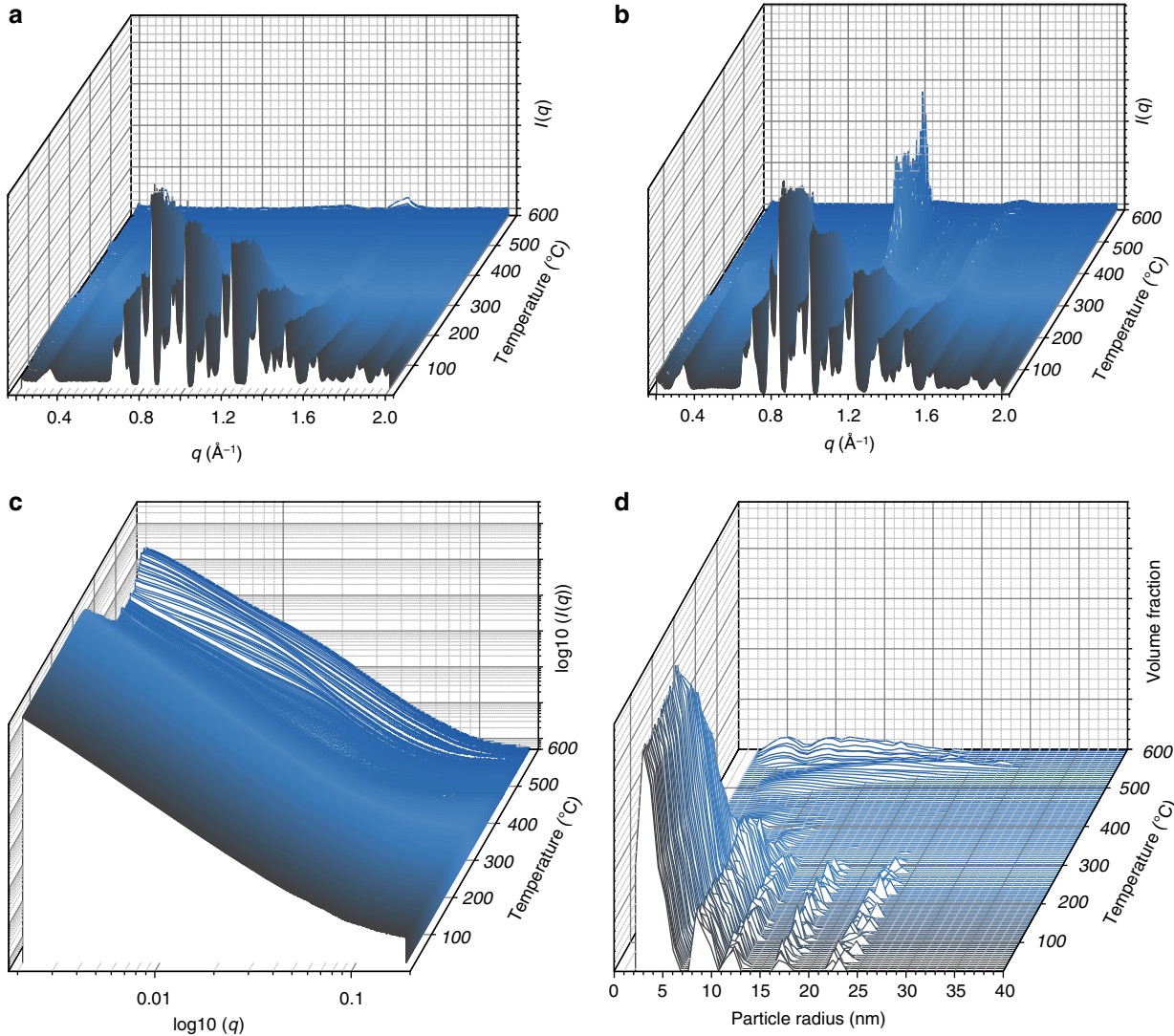

**Fig. 2** Temperature-resolved diffraction. **a** Temperature-resolved WAXS profile of ZIF-62 upon heating from 25 °C to 600 °C. **b** The corresponding data for (ZIF-4-Co)(ZIF-62)(50/50). **c** Temperature-resolved SAXS profile for (ZIF-4-Co)(ZIF-62)(50/50). **d** Temperature-resolved volume fraction distributions of (ZIF-4-Co)(ZIF-62)(50/50)

sample, a second resolved peak of the bIm ligand ($CH$CN$_{bIm}$) can be integrated relative to NC$H$N in Im, giving a 1:0.12 ± 0.01 ratio (expected integration from stoichiometry is 1:0.13), confirming that any loss of ligand in the amorphization process is negligible and/or below the detection limit of NMR. The absence of impurity peaks in the 7–9 ppm region indicates minimal decomposition of imidazolates during digestion/amorphization.

The chemical structure of the blend was probed through synchrotron neutron and X-ray total scattering. Whereas the X-ray structure factor $S(Q)$ of (ZIF-4-Co)(ZIF-62)(50/50) contained Bragg diffraction, that of (ZIF-4-Co)$_{0.5}$(ZIF-62)$_{0.5}$, as expected, did not. This rules out small regions of crystallinity in the latter (Fig. 3a). After appropriate data corrections (see Methods section), the data were converted to the corresponding pair distribution functions (PDFs) (Fig. 3b), which are weighted histograms of the atom pair distances present in both samples. Interatomic distances at 1.3, 2, 3, 4 and 6 Å were common between both crystal and blend samples, consistent with previous conclusions on near-identical short-range order between crystal and glass ZIFs[14].

Above this distance, oscillations at high $r$ were present from the crystalline mixture (ZIF-4-Co)(ZIF-62)(50/50), though the PDF

of (ZIF-4-Co)$_{0.5}$(ZIF-62)$_{0.5}$ was relatively featureless. A dual-phase refinement in PDFGUI[41] of the PDF for (ZIF-4-Co)(ZIF-62)(50/50) was performed in the range 1–15 Å, confirming the presence of both crystalline phases (Fig. 3b inset). Neutron total scattering was also carried out on a deuterated sample of (ZIF-4-Co)$_{0.5}$(ZIF-62)$_{0.5}$ (Supplementary Figures 11 and 12). The expected C-D peak below 1 Å was not visible in the PDF, due to the sample containing a higher hydrogen content than expected. Above this distance and below 6 Å, the PDF was similar to those previously reported for deuterated Zn(Im)$_2$ polymorphs[42].

To probe the evolution in domain structure or size upon heating, synchrotron X-ray diffraction data were collected on a sample of (ZIF-4-Co)$_{0.5}$(ZIF-62)$_{0.5}$ heated from room temperature to 460 °C (Fig. 3c). The first sharp diffraction peak in the $S(Q)$ varied little in intensity or position. While the second and third peaks also remained approximately invariant on heating, some 'flattening' of features at high $Q$ values occurred upon heating above 300 °C. This temperature corresponds to the $T_g$ of (ZIF-4-Co)$_{0.5}$(ZIF-62)$_{0.5}$, and the flattening is consistent with formation of a more liquid-like state. The peak in the $D(r)$ at $r =$ 1.3 Å, which only contains contributions from C-C and C-N pairs

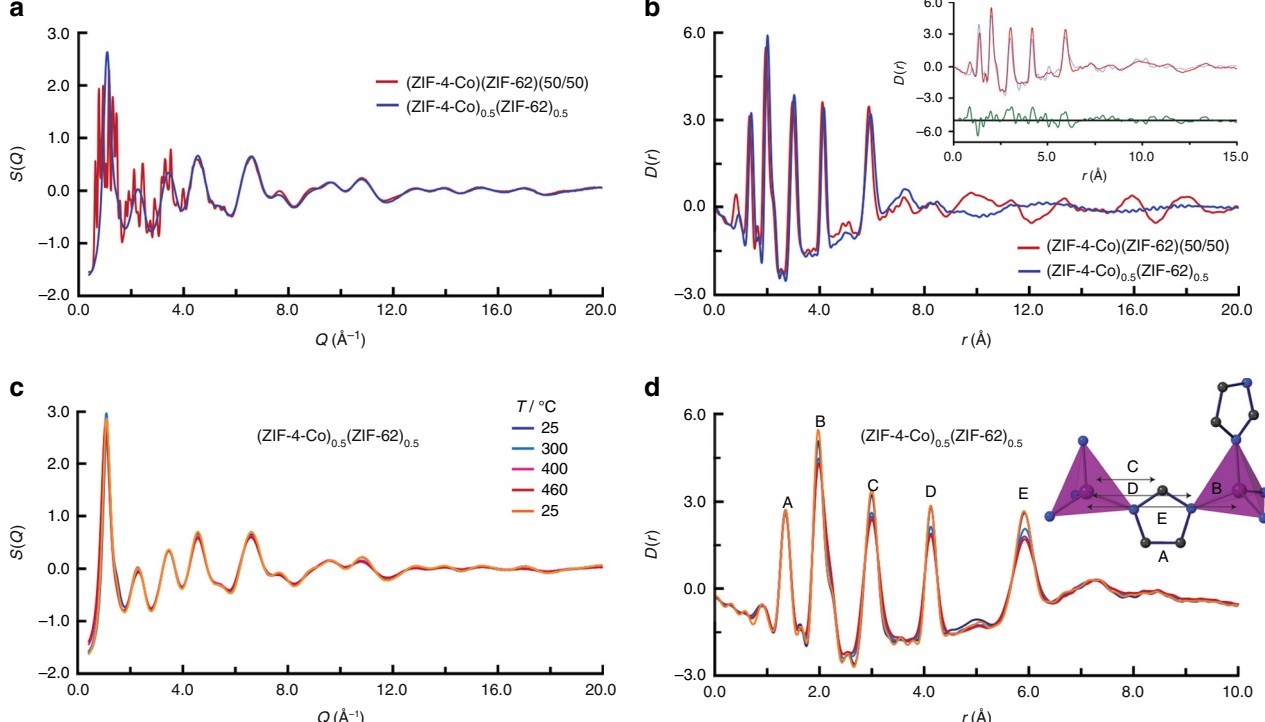

**Fig. 3** Intra-domain structure. **a** X-ray structure factors $S_x(Q)$ of (ZIF-4-Co)(ZIF-62)(50/50) and $(ZIF-4-Co)_{0.5}(ZIF-62)_{0.5}$. **b** Corresponding X-ray pair distribution functions, $D(r)$. Inset: refinement of (ZIF-4-Co)(ZIF-62)(50/50) against the published structure files for ZIF-62 and ZIF-4-Co. Fit—broken blue line. **c** X-ray structure factors of $(ZIF-4-Co)_{0.5}(ZIF-62)_{0.5}$ upon heating. **d** Pair distribution functions $D(r)$ of $(ZIF-4-Co)_{0.5}(ZIF-62)_{0.5}$ upon heating, with the atom pairs that contribute most of the intensity in the labelled peaks indicated in the structural fragment (A–E)

and no contributions from pairs involving Co or Zn, remained constant in intensity and position (Fig. 3d). Those peaks arising mainly from M-$N_1$ ('B'~2 Å), M-C ('C'~3 Å), M-$N_2$ ('D'~4 Å) and M-M ('E'~6 Å) correlations, however, were observed to undergo a reduction in intensity upon heating. The intensity recovered upon cooling back to ambient temperature, showing that no permanent change in short-range order had taken place.

To investigate the suitability of transmission electron microscopy as a characterization technique for MOF glasses, pure samples of crystalline ZIF-62, (ZIF-4-Co)(ZIF-62)(50/50) and $(ZIF-4-Co)_{0.5}(ZIF-62)_{0.5}$ were investigated by electron energy loss spectroscopy (EELS, Fig. 4a, Supplementary Figures 13 and 14). The $K$ (1s) ionization edges for C and N atoms exhibited high-intensity $\pi^\star$ peak features, which are a signature of conjugated heterocycles and consistent with the $\pi^\star$ signature previously reported for EELS of molecular imidazole[43]. These observations demonstrated that the ligands were not damaged under the selected electron beam conditions used. A sample of (ZIF-4-Co)$_{0.5}$(ZIF-62)$_{0.5}$ was subsequently investigated using annular dark field (ADF) scanning transmission electron microscopy (STEM) exhibiting thickness and atomic number contrast and EELS and X-ray energy dispersive spectroscopy (EDS) for chemical mapping at similar or lower electron beam exposures.

EELS performed on a single shard of the glass (Fig. 4a) clearly showed the presence of Co and Zn (Fig. 4b) along with an interfacial region. EDS was performed to yield more insight into the domain structure and interfacial bonding present in the glass particles too thick for EELS analysis (Fig. 4b, c). These revealed a more extended network exhibiting relatively sharp interfaces between Co and Zn domains. Domain sizes were observed ranging from 200 nm to >1 μm in width. This is markedly different to (ZIF-4-Co)(ZIF-62)(50/50), where separate particles of each framework, without domain mixing, were located

(Supplementary Figure 15). In STEM analyses, the electron probe is transmitted through the sample, resulting in EELS and EDS signals that arise from the entire volume through the three-dimensional sample. As a result, these two-dimensional analyses alone were not sufficient to fully characterize the interfaces between the lamellar domains of Co and Zn MOFs. Two-dimensional interface regions with mixed signal composition are not distinguishable from single-phase compositional domains overlapping along the electron beam direction.

EDS tomography was performed in order to address this uncertainty and to characterize the sharpness of the interface between the Co- and Zn-containing regions (Fig. 5). A single-piece shard of $(ZIF-4-Co)_{0.5}(ZIF-62)_{0.5}$ was located that contained two large domains of predominantly Co and Zn, respectively. At the interface, there were two regions (labelled 1 and 2 in Fig. 5) characteristic of heterogeneous mixing between the Co and Zn phases and exhibiting a similar interlocked microstructure as those observed in Fig. 4b, c. Inspection of the tomographic reconstruction volumes at these features revealed that, at feature 1, the Co protrusion is present in a region with negligible Zn content. At feature 2, both Co and Zn were found in the same three-dimensional region, suggesting some minor homogeneous mixing. While some regions of the three-dimensional interface exhibited micro-scale mixing of Co and Zn, the majority were segregated into single-metal domains within an interlocked network microstructure.

**Mechanical properties**. Nanoindentation has previously been used to probe the Young's moduli, $E$, of crystalline and amorphous MOFs[44], though it often results in significant differences between the identified values and those gained from computational studies. The Young's moduli provides a descriptor of the stiffness of a structure under strain and is highly dependent on

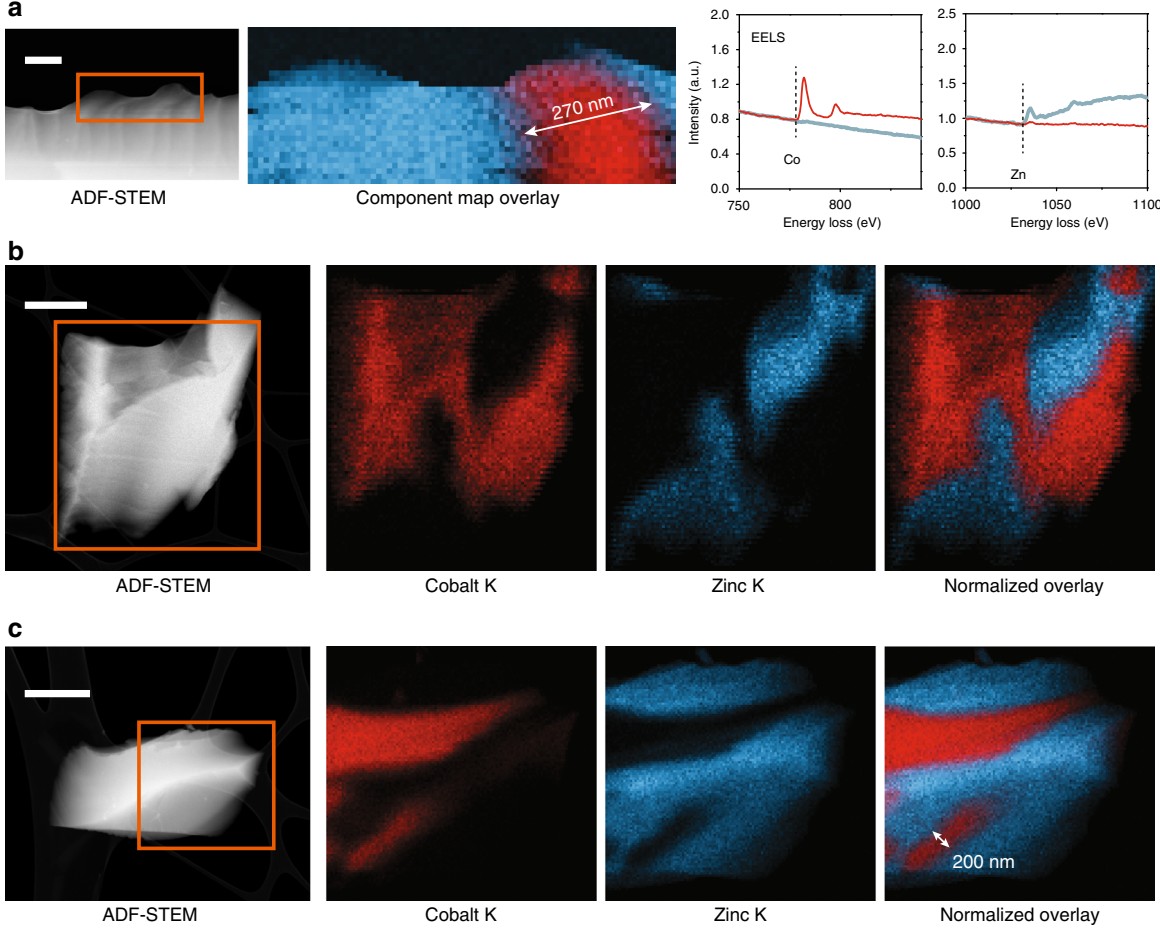

**Fig. 4** Chemical mapping of domain structures in (ZIF-4-Co)$_{0.5}$(ZIF-62)$_{0.5}$. **a** ADF-STEM image and corresponding EELS analysis. Independent component analysis was carried out to separate Co and Zn signals and plotted as a component map overlay (the spectral signals are shown on the right). Scale bar is 250 nm. **b** ADF-STEM image and corresponding X-ray EDS mapping for a second glass particle. Scale bar is 1 μm. **c** ADF-STEM image and corresponding X-ray EDS mapping for a third glass particle. The orange boxes highlight the regions analysed for chemical mapping. Scale bar is 1 μm

the molecular structure. Nanoindentation experiments were thus performed on two independent samples of (ZIF-4-Co)$_{0.5}$(ZIF-62)$_{0.5}$. The existence of both constituent phases in a single glass monolith, in domain sizes smaller than the indenter tip, was confirmed by the consistency across measurements of $E$. Average values of $E$ (7.5 ± 0.5 GPa and 7.1 ± 0.4 GPa, Fig. 6a) were recorded from the load–displacement data (Fig. 6b) of indentations on polished glass monoliths (Fig. 6a inset) between 100 nm and 500 nm. These values lie roughly intermediate between the upper bound of that expected for organic polymers and the lower bound for inorganic glasses (Fig. 6c).

The blend is of comparable pycnometric density to single-phase samples of $a$ZIF-4-Co and the ZIF-62 glass[13, 27], though it exhibits less compliant behaviour under the indenter tip. The increase in $E$ relative to the pure samples ($a$ZIF-4-Co, $E = 6.6$ GPa and ZIF-62 glass, $E = 6.1$ GPa) is ascribed to the isothermal treatment of 2 min above $T_m$, which is necessary for blend formation. This is similar to the increase in $E$ from quenching a ZIF-62 liquid from $T_m$ ($E = 6.6$ GPa) and 572 °C ($E = 8.8$ GPa)[13]. It should also be noted that the relatively poor agreement between calculated and experimental values of $E$ for MOFs has been ascribed to various factors including the large surface effects from small single crystals or monoliths, structural defects and macroscale sample cracking[45]. The prior values of $E$ reported for $a$ZIF-4-Co were gained from non-coalesced single-crystal

samples, and the extent of defects in all three systems has not been the subject of investigation.

## Discussion

We have demonstrated that the two MOF liquids derived from ZIF-4-Zn and ZIF-62 can be blended or alloyed together. The resultant melt-quenched glass shows a single $T_g$, the position of which can be controlled according to the sample composition. The resultant glass structure was probed through electron microscopic measurements on a glass derived from ZIF-4-Co and ZIF-62, finding heterogeneous domain formation. Binding between the domains was investigated using electron tomography, showing regions of homogeneous Co and Zn concentration—indicative of liquid–liquid reactivity. The absence of complete homogeneous mixing is in this instance ascribed to the high viscosity of both molten phases. The interfacial binding of the separate MOF domains to one another is entirely consistent with the observed mechanism of MOF melting, which proceeds via imidazolate dissociation from a M$^{2+}$ centre, and subsequent association of a different imidazolate ligand. We therefore ascribe the domain interlocking mechanism to ligand 'swapping' between the liquid MOF phase and amorphous solid, resulting in the heterometallic MOF glasses shown[14]. Similar heterogeneous structures are also found in SiO$_2$-Al$_2$O$_3$ glasses, where SiO$_2$-rich domains are embedded within Al$_2$O$_3$-rich phases[46].

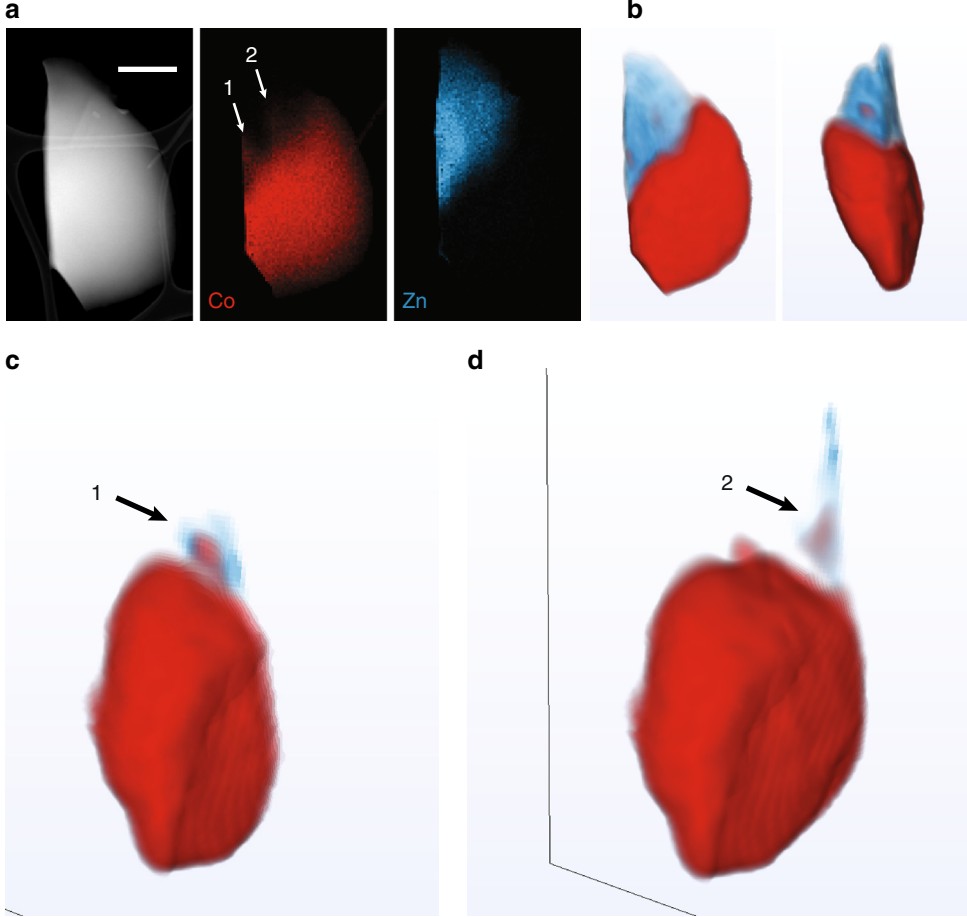

**Fig. 5** EDS tomography of a (ZIF-4-Co)$_{0.5}$(ZIF-62)$_{0.5}$ glass particle. **a** Two-dimensional analyses by ADF-STEM showing the particle morphology and EDS chemical maps of Co and Zn. Scale bar is 500 nm. **b** A volume rendering of the tomographic reconstructions for the Co and Zn signals (two orthogonal viewing directions). **c**, **d** Discrete two-dimensional slices from the three-dimensional volume reconstruction for Zn plotted with the transected volume rendering of the Co reconstruction. Two protrusions from the principal Co domain are highlighted with the numbers 1 and 2. These highlight the extent of three-dimensional spatial overlap in Co and Zn in **c**, **d**

The relationship between domain size and $T_g$ was investigated through the synthesis of two further (ZIF-4-Co)(ZIF-62)(50/50) samples with both larger and smaller initial particle sizes. The first sample was formed through light grinding of ZIF-62 and ZIF-4-Co in a mortar and pestle and compared to a second sample, where the two crystalline frameworks were ball-milled for 20 min together (as opposed to the 5 min used initially). EDS experiments provided qualitative support for the differences in domain sizes of each component using these three different methods of sample preparation (Supplementary Figure 15). Furthermore, DSC experiments confirmed that the sample formed through light grinding contained two distinct $T_g$s (Supplementary Figures 16 and 17), while that formed by ball-milling for a longer time (Supplementary Figure 18) possesses only one, in a near identical position to the original blend sample (ca. 300 ° C).

The binary MOF blend formed and characterized here belongs to the compatible polymer blend category, due to the chemically compatible interactions between the two components and the observation of a single glass transition[47]. A mixture of two chemically incompatible MOF liquids would therefore be expected give rise to an immiscible blend with two or more $T_g$s. The results will prove important in understanding the possibilities afforded by the glass and liquid states of MOFs, demonstrating that blended materials containing two or more MOFs can be produced. We have also shown that the reactivity of the liquid MOF state may be utilized in binding to other MOF components and that the $T_g$ of MOF glasses may be tailored according to blend composition.

## Methods

**Synthesis**. All crystalline samples studied here crystallize in the *Pbca* space group, with cell volumes of 4342 Å$^3$, 4280 Å$^3$ and 4466 Å$^3$ for ZIF-4-Zn, ZIF-4-Co and ZIF-62, respectively. The preparation of mixed samples was done in 0.5 g quantities. For example, for a 50/50 ratio mixture, 0.25 g of each MOF was placed in a 10 ml stainless steel jar, along with 2 × 7 mm diameter stainless steel balls. The mixture was then milled for 5 min (or, to produce a finer particle size for one control sample, for 20 min) in a Retsch MM400 grinder mill operating at 25 Hz. Powder X-ray diffraction patterns of both ball-milled mixtures are shown in Supplementary Information, demonstrating the lack of amorphization.

**Differential scanning calorimetry**. DSC characterizations were conducted using a Netzsch STA 449 F1 instrument in platinum crucibles at a 10 °C min$^{-1}$ heating rate. The simultaneous DSC–thermogravimetric analysis in Supplementary Figure 5 was performed using a TA instruments Q-600 series differential scanning calorimeter, with the sample (~7 mg) held on an aluminium pan under a continuous flow of dry Ar gas. The data were obtained using a heating rate of 10 °C min$^{-1}$. $T_g$s were determined by the method described elsewhere[48].

**X-ray powder diffraction**. Data were collected with a Bruker-AXS D8 diffractometer using Cu Kα ($\lambda = 1.540598$ Å) radiation and a LynxEye position sensitive detector in Bragg–Brentano parafocussing geometry.

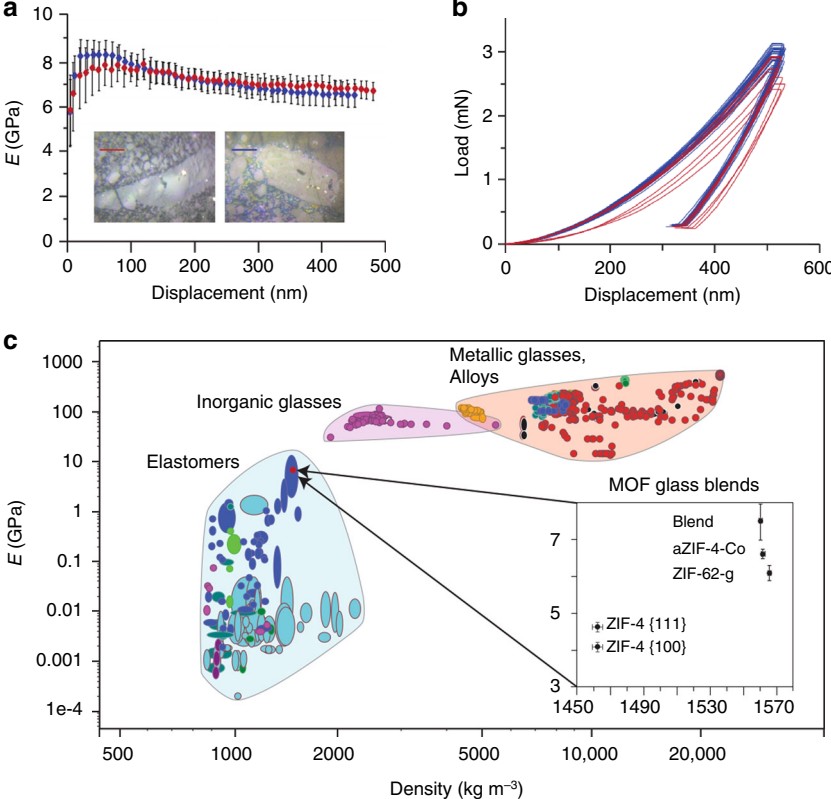

**Fig. 6** Mechanical properties. **a** The Young's modulus, $E$, as a function of indentation depth for two samples of $(ZIF-4-Co)_{0.5}(ZIF-62)_{0.5}$. Error bars represent the standard deviation of 26 measurements for sample 1 (blue) and 8 for sample 2 (red). Inset—optical images of the two samples. Scale bars are 20 µm. **b** Load–displacement curves for both samples and **c** Ashby-style plot of existing alloys, blends and glasses, with the materials here placed into context. Data for ZIF-4 was taken from previous work[27]. Different shadings represent broad material classes

**Combined SAXS and WAXS**. X-ray data were collected at the I22 beamline at the Diamond Light Source, UK ($\lambda = 0.9998$ Å, 12.401 keV). The SAXS detector was positioned at a distance of 9.23634 m from the sample as calibrated using a 100 nm period $Si_3N_4$ grating (Silson, UK), giving a usable $Q$ range of 0.0018–0.18 Å$^{-1}$. The WAXS detector was positioned at a distance of 0.16474 m from the sample as calibrated using a standard $CeO_2$ sample (NIST SRM 674b, Gaithersburg, USA), giving a usable $Q$ range of 0.17–4.9 Å$^{-1}$. Samples were loaded into 1.5 mm diameter borosilicate capillaries under argon inside a glovebox and sealed with Blu-tac and Para-film to prevent the ingress of air. Samples were heated using a Linkam THMS600 capillary stage (Linkam Scientific, UK) from room temperature to 600 °C at 10 °C min$^{-1}$. Simultaneous SAXS/WAXS data were collected every 1 °C. Data were reduced to one dimensional using the DAWN package[49, 50] and standard reduction pipelines[51]. Values for the power law behaviour of the samples were found using the power law model of SASView 4.1.1[52]. Data were fitted over the range $0.003 \leq q \leq 0.005$ Å$^{-1}$. Particle size distributions were calculated using the McSAS package[53, 54], a minimal assumption Monte Carlo method for extracting size distributions from small-angle scattering data. Data were fitted over the range $0.002 \leq q \leq 0.18$ Å$^{-1}$ with a sphere model.

**NMR spectroscopy**. NMR samples were prepared by digesting ~8 mg of sample in 100 µL of 35 wt% DCl in $D_2O$ (purchased from Sigma Aldrich, 99% deuterated) then dissolved in 500 µL of DMSO-$d_6$ (purchased from Sigma Aldrich, 99.9% deuterated). All $^1H$ NMR spectra were recorded on a Bruker Avance III 400 MHz spectrometer.

**Total scattering measurements**. X-ray data were collected at the I15-1 beamline at the Diamond Light Source, UK ($\lambda = 0.161669$ Å, 76.7 keV). A sample of (ZIF-4-Co)(ZIF-62)(50/50), and a small amount of the $(ZIF-4-Co)_{0.5}(ZIF-62)_{0.5}$ sample used in the neutron total scattering experiment were loaded into borosilicate glass capillaries of 1.17 mm (inner) diameter. Data on the samples, empty instrument and capillary were collected in the region of ~0.4 < $Q$ <~ 26 Å$^{-1}$. Corrections for background, multiple scattering, container scattering, Compton scattering and absorption were performed using the GudrunX program[55, 56]. Variable temperature measurements were performed using an identical set-up, though the capillaries were sealed with araldite. Data were taken upon heating at 25 °C, 100 °C, 200 °C, 280 °C and then in 10 °C steps to 340 °C. Further data were collected in 20 °C

intervals to 460 °C, before cooling and a final data set taken at room temperature. Data were corrected using equivalent measurements taken from an empty capillary heated to identical temperatures. Published structures for ZIF-4-Co and ZIF-62 were used to refine data in PDFGUI[10, 33]. The values $U_{11}$, $U_{22}$ and $U_{33}$ were set to 0.003 Å$^2$ and constrained to be isotropic. Cross-diagonal terms were set to 0, and data beyond 15 Å were not fitted because of the lack of intensity. The final $R_w$ value was 0.34, due to some disordering in the initial mixture introduced by ball-milling.

Deuterated samples of ZIF-4-Co and ZIF-62 were prepared by equimolar replacement of the hydrogenated benzimidazole and imidazole in their respective syntheses, by the deuterated equivalents, supplied by the ISIS Deuteration Facility. A glass sample of $(ZIF-4-Co)_{0.5}(ZIF-62)_{0.5}$ was then produced as reported in the manuscript. Data were measured at room temperature using the NIMROD diffractometer at ISIS[57]. A sample was placed into an 8-mm diameter thin-walled vanadium can, and data from an empty vanadium can, empty instrument, 8 mm V-5.14% Nb rod was used to correct the data in the Gudrun software[55].

**Gas pycnometry**. Pycnometric measurements were carried out using a Micromeritics Accupyc 1340 helium pycnometer. The typical mass used was 200 mg; the values quoted are the mean and standard deviation from a cycle of 10 measurements.

**Electron microscopy and spectroscopy**. STEM data were acquired using an FEI Osiris microscope equipped with a high-brightness X-FEG electron source and operated at 80 kV. The beam convergence was set to 11.0 mrad. EELS was acquired using a post-column Gatan Enfinium spectrometer. A 2.5 mm entrance aperture was selected, defining a collection semiangle of 19.4 mrad. Spectra were acquired in dual EELS mode: electrons undergoing no inelastic scattering (the zero loss peak) and those undergoing low energy losses were recorded with a fast acquisition time (0.0001 s) and nearly simultaneously electrons undergoing inelastic scattering at element-specific core loss ionization edges were recorded at longer exposures times (100 ms exposure at C and N $K$ edges and 500 ms at Co and Zn $L_{23}$ edges). Probe currents in this electron optical configuration were typically <150 pA. X-ray EDS was acquired using a 'Super-X' EDS detector system with four detectors mounted symmetrically about the optic axis of the microscope (200 ms per pixel). For all spectroscopic data, images were also simultaneously recorded on ADF detectors. These images contain atomic number and thickness contrast, giving structural

information in parallel with the chemical mapping obtained in the EELS and EDS data. For EDS tilt-series tomography, EDS spectrum images were acquired from −70° to 70° in 10° increments.

Data were processed using Hyperspy[58], an open-source software coded in Python. For EELS data, the spectra were first aligned to the ZLP, initially by shifting the maximum intensity channel to zero followed by cross-correlation-based sub-pixel alignment. Spikes due to X-rays striking the charge-coupled device detector were removed using a routine that automatically identified outlying high-intensity pixels and performed interpolation in the spectral region after removal of the spike. Independent component analysis was likewise performed in Hyperspy. For tilt-series tomography, Zn and Co chemical maps were initially combined for alignment of the tilt-series image-stack. In order to correct for detector shadowing as a function of tilt angle, the chemical maps were re-normalized to maintain constant integrated intensities at all tilts. This procedure was based on the constant total quantity of Zn and Co in the particle recorded within the field of view at all tilt angles. The combined Zn and Co image-stack was aligned using Scikit-Image, an open-source image processing software coded in Python, first using cross-correlation, and then the tilt axis was subsequently aligned by applying shifts and rotations to minimize artefacts in back projection reconstructions. The alignments were then applied to each of the Zn and Co tilt series. A compressed sensing reconstruction algorithm coded in MATLAB (Mathworks) was then used to perform the final independent reconstructions of the Zn and Co tilt series. Broadly, compressed sensing tomography approaches make use of prior knowledge of the sparsity of the signal undergoing reconstruction in a particular transform domain (the sparsity is given as the number of non-zero intensities) in order to recover high-fidelity tomographic reconstructions from highly undersampled tilt-series data[59, 60]. This compressed sensing tomography implementation used three-dimensional total generalized variation[61] regularization for the sparsity constraint in conjunction with a real-space projection operator from the Astra toolbox[62] and using the primal-dual hybrid gradient method[63] to solve the reconstruction problem. Reconstructions were further processed in ImageJ and FEI Avizo software for visualization. The total particle shape recovered in the tomographic reconstruction was used to threshold the volume to remove spurious signals due to noise in the reconstruction volume outside the particle sub-volume. No further processing was applied to the intensities within the particle.

**Nanoindentation.** The Young's modulus (*E*) of the samples was measured using an MTS Nanoindenter XP at ambient conditions. Samples were mounted in an epoxy resin and polished using increasingly fine diamond suspensions. Indentation experiments were performed under the dynamic displacement controlled mode, at a constant strain rate of $0.05\,s^{-1}$. All tests were conducted using a three-sided pyramidal (Berkovich) diamond indenter tip, to a maximum surface penetration depth of 500 nm. The load–displacement data collected were analysed using the Oliver and Pharr method[64]. A Poisson's ratio of 0.2 was used, in accordance with prior studies on ZIF materials[65].

**Data availability.** The data that support the findings of this study are available from the corresponding author upon request.

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

## Acknowledgements

T.D.B. would like to thank the Royal Society for a University Research Fellowship, and L. L. and N.J.B. acknowledge EPSRC studentships. C.W.A. acknowledges the Royal Society for funding. C.Z. acknowledges the financial support from China Scholarship Council and the Elite Research Travel Scholarship from the Danish Ministry of Higher Education and Science. We acknowledge the provision of synchrotron access to beamlines I15-1 (EE171151) and I22 (NT18236-1) at the Diamond Light Source, Rutherford Appleton Laboratory UK. The authors thank Jiayan Zhang for assistance for a DSC measurement. This work benefited from the use of the SasView application, originally developed under NSF Award DMR-0520547. SasView also contains code developed with funding from the EU Horizon 2020 programme under the SINE2020 project Grant No. 654000. Experiments at the ISIS Pulsed Neutron and Muon Source (on the NIMROD instrument) were supported by a beam-time allocation from the Science and Technology Facilities Council, on proposal RB1720006. S.M.C. acknowledges the Henslow Research Fellowship and Girton College, Cambridge. S.M.C. and P.A.M. acknowledge funding from the European Research Council under the European Union's Seventh Framework Program (No. FP7/2007-2013)/ERC Grant Agreement No. 291522-3DIMAGE. C.-B.S. acknowledges support from the Leverhulme Trust project Breaking the non-convexity barrier, EPSRC grant EP/M00483X/1, EPSRC centre EP/N014588/1 and from CHiPS (Horizon 2020 RISE project grant). R.T. acknowledges funding from EPSRC grant EP/L016516/1 for the Cambridge Centre for Analysis. R.T. and C.-B.S. also acknowledge the Cantab Capital Institute for the Mathematics of Information.

## Author contributions

T.D.B. designed the project and wrote the manuscript with L.L. and S.M.C., with input from all authors. Electron microscopy was performed by P.A.M. and S.M.C. SAXS/WAXS experiments were performed by T.D.B., A.J.S., N.T. and G.J.S., with analysis performed by A.J.S. and G.J.S. DSC measurements and sample preparation were carried out by L.L. and C.Z., facilitated by Y.Y. Deuterated organic ligands were provided by S.E. N., and PDF measurements carried out and analysed by T.D.B., D.A.K., C.W.A., C.Z., L. L., T.F.H. and P.A.C. NMR measurements were carried out and analysed by F.B. and N.J. B. R.T. and C.-B.S. developed the tomography reconstruction code.

## Additional information

**Competing interests:** The authors declare no competing interests.

