## [Peer Review File · Nature Communications]

Reviewers' comments:

Reviewer #1 (Remarks to the Author):

This is an interesting paper aimed at demonstrating the possibilities associated with 'blending' metal-organic frameworks into new galley materials. Overall, I think that the idea that MOFs can be stable when molten and can then be blended is an intriguing idea that deserves to be published. The authors have worked very hard to characterise their materials using an array of techniques, and I think this adds to the strength of the paper. Therefore I am happy to recommend that the paper be published subject to a few changes.

1. In some places I found the paper quite difficult to follow. For example, in the first paragraph at the top of page 4, after the sentence

"Accordingly, the formation of the liquid and glass phases of MOFs has recently emerged as a new area in an ever- expanding field."

The paragraph then goes on to talk about the 'former' and the 'latter' but doesn't seem to make much sense as the former seems to refer to glass and the latter to crystalline solids. I couldn't follow what the authors really wanted to communicate. This is not the only example - there are other paragraphs that I had to read several times and even then I didn't really get what was being written. I suggest that the authors, as predominantly native English speakers, take a little time to carefully read and correct every paragraph to make sure their intended meaning is clear.

2. The samples show 'mass additivity behaviour' similar to other quoted polymer blends. Does this property indicate anything in particular about the chemistry going on at the interface. Does the size of the domains have any influence on this property (presumably if the domains were large enough you would see two Tg temperatures in the DSC - what controls when one Tg is seen even when there are almost micron-sized domains?

3. If I understand correctly the Co version of the MOF doesn't actually melt, but simply becomes amorphous and then recrystallises (into a different structure?). Is this correct? Given that the domain size of the final blend is important this will presumably be an important feature (as the authors state that viscosity is important in not getting full miscibility). Would the structure of the fully zinc material be different in this regards because there should be better miscibility of the molten MOFs.

4. Top of page 9, there is sentence which refers to 'difference internal pore structures'. Is there a word missing? Other it makes no sense.

5. In the liquid NMR section, since some of the work talks about liquid MOFs (to an extent) I think it would be clearer if the results sections says exactly what the MOF blend has been digested in - just to avoid any possibility of confusion.

Russell Morris

Reviewer #2 (Remarks to the Author):

The paper describes the preparation of Zn-Zn-ZIF or Co-Zn-ZIF alloy glasses by melt-quench techniques. The authors reported the thermal properties, crystallographic structural characterization by X-ray and neutron diffraction, and the distribution of Co and Zn domains by element mapping of electron tomography. Each characterization was carefully conducted with high reliability. On the other hand, all the single phase compounds in this work are reported previously

by the same research group.

The previous reports (ref 13, 14, 21, etc.) discussed the characterization including PDF, SAXS, DSC, neutron scattering, indentation for ZIF glass and the techniques were also used for this study. Overall, it is not easy to find the novelty on the materials and characterization, and thus new phenomena should be found for the blended glass by melt-quenching in terms of glass-functionality of structure.

In the preparation of blend sample, they heated the mixture samples of Zn-MOF and Co-MOF where one of them melts, and the (presumably) liquid-solid mixture was quenched to have a glass state. It is reasonable to find the mixture glass of them.

The reviewer would expect new results in this work. However, the blended MOF glasses having unique functionality, thermal behavior, mechanical properties are not observed in their parent phases. They do not study on the functionality of the glass. Mechanical properties (Figure 6) for the blend sample show fairly similar density and Young's modulus to those of their parent phases and no advantages are observed. One significant data is the demonstration of systematic tuning of T_g (Figure 1c) by changing the ratio of $(\text{ZIF-4-Zn})_{1-x}(\text{ZIF-62})_x$. The T_g changes linearly as a function of x , and this is the first example to control T_g values of MOF glasses.

The study of T_g in Figure 1c and the results of electron tomography for $(\text{ZIF-4-Co})_{0.5}(\text{ZIF-62})_{0.5}$ (Figure 4, 5) arise a question about the correlation between the domain size of each MOF and the resultant T_g , but the present work does not explain the relationship. The glass transition is correlated with the relaxation of some domain in the MOF structure. The important thing is whether we could observe a dependence of T_g on the domain size, the 200 nm to $> 1 \mu\text{m}$, by the tomography (Figure 4 and 5). This is not convincing.

My feeling is that the regime of 200 nm to $> 1 \mu\text{m}$ is quite large and it is hard to say "blended", even though they claim the interface of each domain could have homogeneous mixing part. Mechanical properties by the indentation measurement also depend on the selected surface of such large-domain "blend". If high viscosity and poor miscibility are the reason to have the segregated domain as they claim, the authors should do more efforts to have homogeneous blending, but no control experiments to change the domain distribution are observed. Again, the large regime of each domain observed in $(\text{ZIF-4-Co})_{0.5}(\text{ZIF-62})_{0.5}$ (and $(\text{ZIF-4-Zn})_{1-x}(\text{ZIF-62})_x$) is not obviously correlated with the T_g and the discussion about the blending to render different properties of MOF glasses is weak in the current form.

Because of the reasons of lacking novelty on materials, functions, unique properties on "blended glass" and poor tunability of blending of two MOFs, I could not recommend the present manuscript for publication in Nature Communication.

Reviewer #3 (Remarks to the Author):

Longley et al. explore the fabrication of hybrid glass blends through melting mixtures of ZIF-4 and ZIF-62 mesoporous MOFs, subsequently termed "MOF blends". Their study follows-up on the earlier introduction of hybrid glasses by the same authors – a novel, prolific type of glassy materials which is obtained through melt-quenching certain MOF materials.

On a general note, I am in favor of publication of this article in Nature Communications. I feel that the presented results and, in particular, the concept of hybrid glass blends will be of broad interest to the community, and will trigger significant work towards the integration of actual functions into this new class of glasses.

The most interesting part of the study is the underlying concept, where common understanding of traditional glasses is transferred so as to significantly widen the emerging field of hybrid glasses.

That is, one of the most important topo-chemical tools in classical glass chemistry is the mixing of cation and anion species or super-structural units, creating mixed glass networks. Here, such mixing has become key in the design of complex properties as well as in tailoring processability, for example, through adjusting the temperature dependence of viscosity or cation mobility.

Longley et al. demonstrate in a convincing example that mixed networks can indeed be fabricated through melting mixtures of two MOF species. In my opinion, this evidence lies primarily in the

TEM tomographic analyses of the interface regime which occurs between the observed chemical domains. While the overall microstructure is probably a relict of the grain or crystallite sizes of the mixed species (it would be helpful if the authors could provide these values in the methods-section), shorter-range intra-domain structural analyses (especially SAXS) and the interface gradient confirm actual network mixing. (it might be noteworthy that similar long-range domain structures are regularly observed in other particle-derived glasses, for example, SiO₂-Al₂O₃ or SiO₂-TiO₂ in which, however, network mixing is still observed on shorter length scale).

From the observed domain structure, personally, I would still expect two Tgs rather than a single value. Eventually, these two Tgs are overlapping to a large extent so that they are not readily visible by DSC at 10 K/min. Perhaps DSC at other heating rates could help, or a more sensitive analysis of internal friction through DMA. However, I fully understand that the latter is strongly limited at present due to the shape and size of samples. Considering Fig. S2, I recommend to the authors to elaborate a little bit more on the presented data and how Tgs were extracted. Clearly, Tg of ZIF-4 is visible only as a shallow kink; similar kinks occur on other regions of the curve, for example, at 276 °C. It might be worthwhile to comment on the sensitivity and reproducibility of the instrument employed by the authors in the considered temperature range. In the method section, the authors state that Cp analyses were performed. However, throughout the paper, they do not show Cp data, but focus on heat flow in a.u. or in W/g. This should be commented on or harmonized. From the scattering on the curve in S2, I assume that it is shown at higher magnification than the curves in S3 or 1b. Perhaps it would be helpful to the reader to have a zoom at the ~ 300 °C region in Figure 1b for better comparability? Also, the assignment of Tg is somewhat unclear: in Figure 1b, the authors indicate the peak at ~ 330 °C (blue curve), in Figure S3, they use the onset of this peak (and probably derive Figure 1c from this), and in Figure S2, they use two shallow local minima (why not the peak at ~ 340 °C?). All this should be harmonized or explained more clearly.

The authors also conduct nanoindentation experiments, from which they draw Young's modulus. Noteworthy to the reader, nanoindentation may cause significant experimental error (overestimation) on E because of erroneous approximation of the indentation contact area (for example, caused by pile-up, sink-in or other size-dependent effects). This puts some uncertainty into Figure 6c, in addition to the error bars which are obtained from experimental statistics. On the other hand, this does not – in my opinion – affect the overall results of the paper or my recommendation to publish, since the observed trend is certainly correct. I wonder whether the authors could add data on pristine ZIF-62 to the inset of Figure 6c.

As another minor point, I suggest to use the term "tailoring" instead of "tuning" in the context of Tg adjustments.

Response to Referees: Liquid Phase Blending of Metal-Organic Frameworks

We would like to thank you and the three referees, for the time taken in considering this manuscript. We have read the comments of the referees, and performed the major revisions requested. Extra synthesis, electron microscopy and differential scanning calorimetry experiments were carried out, along with the requisite analysis. Changes have been highlighted in yellow in the manuscript, and our responses (in black) to the original comments (in blue) below. We hope that this will prove suitable for publication and look forward to hearing from you.

Referee 1

This is an interesting paper aimed at demonstrating the possibilities associated with 'blending' metal-organic frameworks into new galley materials. Overall, I think that the idea that MOFs can be stable when molten and can then be blended is an intriguing idea that deserves to be published. The authors have worked very hard to characterise their materials using an array of techniques, and I think this adds to the strength of the paper. Therefore, I am happy to recommend that the paper be published subject to a few changes.

We are delighted that the referee believes this paper should be published, and thank them very much for their support – we too are very excited about the possibilities for new materials using this method.

In some places I found the paper quite difficult to follow. For example, in the first paragraph at the top of page 4, after the sentence "Accordingly, the formation of the liquid and glass phases of MOFs has recently emerged as a

new area in an ever-expanding field. "The paragraph then goes on to talk about the 'former' and the 'latter' but doesn't seem to make much sense as the former seems to refer to glass and the latter to crystalline solids. I couldn't follow what the authors really wanted to communicate. This is not the only example - there are other paragraphs that I had to read several times and even then I didn't really get what was being written. I suggest that the authors, as predominantly native English speakers, take a little time to carefully read and correct every paragraph to make sure their intended meaning is clear.

The advice of the referee is greatly appreciated – we have re-read the manuscript and clarified the instances where we think the manuscript needed improving. For example, in the 'former' and the 'latter' case mentioned by the referee above, the sample names have been included to make it clear what we are referring to. We have also tried to make the text 'flow' more naturally, and this results in a much improved manuscript.

The samples show 'mass additivity behaviour' similar to other quoted polymer blends. Does this property indicate anything in particular about the chemistry going on at the interface. Does the size of the domains have any influence on this property (presumably if the domains were large enough you would see two T_g temperatures in the DSC - what controls when one T_g is seen even when there are almost micron-sized domains?

This is a dimension to the research study that, admittedly, did not occur to us. All three referees are interested in the relationship between domain size and T_g , and so we have performed additional experiments (Figs. S12, S13) to investigate this for the revised manuscript, and included text in the discussion section. Specifically, we concentrated on the effect of (i) very large and (ii) very small domain sizes upon T_g . In the original manuscript, a sample of (ZIF-4-Co)(ZIF-62)(50/50) was prepared by ball-milling the two crystalline phases together for 5 minutes. A sample (ZIF-4-Co)_{0.5}(ZIF-62)_{0.5} was then prepared by melt-quenching. In our revised experiments, we have prepared further samples of (ZIF-4-Co)(ZIF-62)(50/50) by (i) ball-milling ZIF-4-Co and ZIF-62 for 20 minutes (i.e. to reduce particle size further), and (ii) by grinding in a mortar and pestle without any ball-milling (i.e. to try and maximise particle size). DSC and SEM experiments were carried out in each case, and conclusions drawn upon the influence of domain size upon melting. We have confirmed the astute prediction of the reviewer, i.e. that in the case of the hand ground sample with larger domain sizes, two glass transitions are evident, whilst in the case of the ball-milled sample with smaller domain sizes, only one T_g remains.

If I understand correctly the Co version of the MOF doesn't actually melt, but simply becomes amorphous and then recrystallises (into a different structure?). Is this correct? Given that the domain size of the final blend is important this will presumably be an important feature (as the authors state that viscosity is important in not getting full miscibility). Would the structure of the fully zinc material be different in this regards because there should be better miscibility of the molten MOFs.

The referee is correct in his understanding. The text on page 6 has been changed slightly to make this clear. We also recognise that the domain structure of the fully zinc material may well be different – though encouragingly both materials (the Co/Zn blend and the fully Zn one) only possess one glass transition in the differential scanning calorimetry measurements. However, computational estimates of the viscosity of Zn-based ZIF-4 (Gaillac, R., et al., *Liquid Metal-Organic Frameworks*. Nature Materials, 2017. **16**: p. 1149-1154), and an experimental report of the viscosity of Zn based ZIF-62 (Qiao, A., et al., *A metal-organic framework with ultrahigh glass-forming ability*. Science Advances, 2018. **4**: p. eaao6827.), suggest both are very high and so any differences with the amorphous solid Co-ZIF-4 may be small. We have added some discussion on page 6 to this effect.

Top of page 9, there is sentence which refers to 'difference internal pore structures'. Is there a word missing? Other it makes no sense.

This is a typo and should read "different" not "difference".

In the liquid NMR section, since some of the work talks about liquid MOFs (to an extent) I think it would be clearer if the results sections says exactly what the MOF blend has been digested in - just to avoid any possibility of confusion.

This has been done – we agree that this was confusing.

Referee 2

The paper describes the preparation of Zn-Zn-ZIF or Co-Zn-ZIF alloy glasses by melt-quench techniques. The authors reported the thermal properties, crystallographic structural characterization by X-ray and neutron diffraction, and the distribution of Co and Zn domains by element mapping of electron tomography. Each characterization was carefully conducted with high reliability. On the other hand, all the single phase compounds in this work are reported previously by the same research group.

We thank the referee for their time in reading this manuscript, and for their comments. We are happy that the referee, like referee 1 and 3, has noted the careful characterization of the MOF materials, using multiple techniques.

The previous reports (ref 13, 14, 21, etc.) discussed the characterization including PDF, SAXS, DSC, neutron scattering, indentation for ZIF glass and the techniques were also used for this study. Overall, it is not easy to find the novelty on the materials and characterization, and thus new phenomena should be found for the blended glass by melt-quenching in terms of glass-functionality of structure. In the preparation of blend sample, they heated the mixture samples of Zn-MOF and Co-MOF where one of them melts, and the (presumably) liquid-solid mixture was quenched to have a glass state. It is reasonable to find the mixture glass of them. The reviewer would expect new results in this work. However, the blended MOF glasses having unique functionality, thermal behavior, mechanical properties are not observed in their parent phases. They do not study on the functionality of the glass. Mechanical properties (Figure 6) for the blend sample show fairly similar density and Young's modulus to those of their parent phases and no advantages are observed.

We are happy that the referee agrees with our motivation for investigating the structure of the properties of MOF-glass blends. This paper is notably different from the others the referee mentions, in terms of the materials it characterizes, and, as referee 1 and 3 point out, the novel and interesting idea of blending MOFs. We do not study the functionality of the glass in this work as this lies outside the scope of this proof of concept, synthesis and characterization paper.

One significant data is the demonstration of systematic tuning of T_g (Figure 1c) by changing the ratio of $(ZIF-4-Zn)_{1-x}(ZIF-62)_x$. The T_g changes linearly as a function of x , and this is the first example to control T_g values of MOF glasses. The study of T_g in Figure 1c and the results of electron tomography for $(ZIF-4-Co)_{0.5}(ZIF-62)_{0.5}$ (Figure 4, 5) arise a question about the correlation between the domain size of each MOF and the resultant T_g , but the present work does not explain the relationship. The glass transition is correlated with the relaxation of some domain in the MOF structure. The important thing is whether we could observe a dependence of T_g on the domain size, the 200 nm to $> 1 \mu m$, by the tomography (Figure 4 and 5). This is not convincing. My feeling is that the regime of 200 nm to $> 1 \mu m$ is quite large and it is hard to say "blended", even though they claim the interface of each domain could have homogeneous mixing part. Mechanical properties by the indentation measurement also depend on the selected surface of such large-domain "blend". If high viscosity and poor miscibility are the reason to have the segregated domain as the they claim, the authors should do more efforts to have homogeneous blending, but no control experiments to change the domain distribution are observed. Again, the large regime of each domain

observed in $(\text{ZIF-4-Co})_{0.5}(\text{ZIF-62})_{0.5}$ (and $(\text{ZIF-4-Zn})_{1-x}(\text{ZIF-62})_x$) is not obviously correlated with the T_g and the discussion about the blending to render different properties of MOF glasses is weak in the current form.

The referee, like referee 1, is interested in the relationship between domain size and T_g . Accordingly, this is where we have focused our efforts in revising the manuscript. Please see the additional work noted in the response to referee 1 above.

Because of the reasons of lacking novelty on materials, functions, unique properties on "blended glass" and poor tunability of blending of two MOFs, I could not recommend the present manuscript for publication in Nature Communication.

We hope that the revised version, along with extra experiments linking domain size and glass transition, will to allay some of the referees concerns. However, we (very respectfully, given the time the reviewer has taken to give us advice on the manuscript), disagree that unique properties need to be demonstrated by the glass shown here. In terms of fundamental science, as noted by the other two referees, the idea and the synthesized materials are novel and are the main attractions of the manuscript. All three referees commend the completeness of the characterization. We do however agree with the referee that finding unique properties of the hybrid glasses will be important in the future, and we look forward to the research that ourselves, and others, will do in this area.

Referee 3

Longley et al. explore the fabrication of hybrid glass blends through melting mixtures of ZIF-4 and ZIF-62 mesoporous MOFs, subsequently termed "MOF blends". Their study follows-up on the earlier introduction of hybrid glasses by the same authors – a novel, prolific type of glassy materials which is obtained through melt-quenching certain MOF materials. On a general note, I am in favor of publication of this article in Nature Communications. I feel that the presented results and, in particular, the concept of hybrid glass blends will be of broad interest to the community, and will trigger significant work towards the integration of actual functions into this new class of glasses. The most interesting part of the study is the underlying concept, where common understanding of traditional glasses is transferred so as to significantly widen the emerging field of hybrid glasses. That is, one of the most important topo-chemical tools in classical glass chemistry is the mixing of cation and anion species or super-structural units, creating mixed glass networks. Here, such mixing has become key in the design of complex properties as well as in tailoring processability, for example, through adjusting the temperature dependence of viscosity or cation mobility.

We are delighted that the referee finds this study of considerable interest, in terms of the fundamental science and the idea behind it. We are thankful for the time invested by the referee in reviewing this manuscript.

Longley et al. demonstrate in a convincing example that mixed networks can indeed be fabricated through melting mixtures of two MOF species. In my opinion, this evidence lies primarily in the TEM tomographic analyses of the interface regime which occurs between the observed chemical domains. While the overall microstructure is probably a relict of the grain or crystallite sizes of the mixed species (it would be helpful if the authors could provide these values in the methods-section), shorter-range intra-domain structural analyses (especially SAXS) and the interface gradient confirm actual network mixing. (it might be noteworthy that similar long-range domain structures are regularly observed in other particle-derived glasses, for example, $\text{SiO}_2\text{-Al}_2\text{O}_3$ or $\text{SiO}_2\text{-TiO}_2$ in which, however, network mixing is still observed on shorter length scale).

We agree that the main evidence for the mixed networks lies in the STEM images. However, the secondary evidence is the appearance of the predominant occurrence of a single glass transition event. We also believe that there is a

certain degree (not dominant) of overlapping of the T_g values for both ZIF-4 and -62. Due to the ultrahigh viscosity of ZIF-62 above T_m ,^[1] the diffusion of the structural species is so slow that the middle of the original particles is still dominantly occupied by one single component (ZIF-4 or ZIF-62). In other words, the blend still has a certain degree of microscopic heterogeneity after melting of the mixture.

We are very grateful for the author lending their experience in the wider glass domain to us, and the extremely helpful comment placing this work into context with particle-derived oxide glasses. We have added content in the discussion section to this effect. We have also performed the additional electron microscopy experiments on the physical mixture of ZIF-4-Co and ZIF-62 (Fig. S12), as suggested by the referee, and included these in the SI.

From the observed domain structure, personally, I would still expect two T_g s rather than a single value. Eventually, these two T_g s are overlapping to a large extent so that they are not readily visible by DSC at 10 K/min. Perhaps DSC at other heating rates could help, or a more sensitive analysis of internal friction through DMA.

However, I fully understand that the latter is strongly limited at present due to the shape and size of samples.

We have taken the advice of the reviewer, and have performed additional DSC experiments on the $(\text{ZIF-4-Zn})_{0.5}(\text{ZIF-62})_{0.5}$ blend at a heating rate of 5 °C/min (Fig S6b), to try and answer the question posed. Unfortunately, DMA is not possible on the samples because of the sample morphology, as predicted by the referee. Only one glass transition appears to be present. A full study of the effect of domain structure would be extremely interesting, though, given the length and focus of this paper, perhaps best suited to a follow up investigation.

Considering Fig. S2, I recommend to the authors to elaborate a little bit more on the presented data and how T_g s were extracted. Clearly, T_g of ZIF-4 is visible only as a shallow kink; similar kinks occur on other regions of the curve, for example, at 276 °C. It might be worthwhile to comment on the sensitivity and reproducibility of the instrument employed by the authors in the considered temperature range.

The measurements here require sensitive instruments, though we have measured the T_g of ZIF-4 multiple times,^[2, 3] and found that it is always located in the temperature range of 290-300 °C at a heating rate of 10 K/min. Hence we consider the T_g be at the position (298 °C) in Fig. S2.

In the method section, the authors state that C_p analyses were performed. However, throughout the paper, they do not show C_p data, but focus on heat flow in a.u. or in W/g. This should be commented on or harmonized.

We have corrected this in the methods section and removed the reference to C_p data.

From the scattering on the curve in S2, I assume that it is shown at higher magnification than the curves in S3 or 1b. Perhaps it would be helpful to the reader to have a zoom at the ~ 300 °C region in Figure 1b for better comparability?

This is a good idea, however given the crowded nature of Fig. 1b, we have included this underneath Fig. S2.

Also, the assignment of T_g is somewhat unclear: in Figure 1b, the authors indicate the peak at ~ 330 °C (blue curve), in Figure S3, they use the onset of this peak (and probably derive Figure 1c from this), and in Figure S2, they use two shallow local minima (why not the peak at ~ 340 °C?). All this should be harmonized or explained.

The onset of the peak has been used in all cases – we have changed Fig. 1b to reflect this.

The authors also conduct nanoindentation experiments, from which they draw Young's modulus. Noteworthy to the reader, nanoindentation may cause significant experimental error (overestimation) on E because of erroneous approximation of the indentation contact area (for example, caused by pile-up, sink-in or other size-dependent effects). This puts some uncertainty into Figure 6c, in addition to the error bars which are obtained from experimental statistics. On the other hand, this does not – in my opinion – affect the overall results of the paper or my recommendation to publish, since the observed trend is certainly correct. I wonder whether the authors could add data on pristine ZIF-62 to the inset of Figure 6c.

We agree with the referee on the significant experimental error and overestimation in E , and have added a comment in the manuscript to this effect. Regretfully, we cannot synthesize single crystals of ZIF-62 large enough to provide any meaningful indentation data. The *J. Am. Chem. Soc.* article in which we first published the melting behaviour of ZIF-62 (10.1021/jacs.5b13220) contains our best efforts at these synthesis attempts in the SI. The obtained crystals were well below 50 μm in diameter. We do however see the value in including data on pristine ZIF-4 in the inset of Figure 6c, and have done so accordingly.

As another minor point, I suggest to use the term “tailoring” instead of “tuning” in the context of T_g adjustments.

We have accepted the suggestion of the reviewer, and changed this when mentioning the adjustment in T_g .

References

1. Qiao, A., et al., *A metal-organic framework with ultrahigh glass-forming ability*. *Science Advances*, 2018. **4**: p. eaao6827.
2. Bennett, T.D., et al., *Hybrid glasses from strong and fragile metal-organic framework liquids*. *Nature Communications*, 2015. **6**: p. 8079.
3. Bennett, T.D., et al., *Melt-Quenched Glasses of Metal-Organic Frameworks*. *Journal of the American Chemical Society*, 2016. **138**: p. 3484-3492.

REVIEWERS' COMMENTS:

Reviewer #1 (Remarks to the Author):

Reading through the revised manuscript I believe that the majority of the referee comments have been addressed satisfactorily. I think the blending aspect of MOF glasses constitutes enough novelty for publication.

The manuscript still contains a few errors that should have been cleaned up by now. For example, in the synthesis part of the methods sections, some volumes are listed with units of Angstroms (should of course be cubic Angstroms) and a few lines later there is a reference to "...milled for 5 (or 20?) minutes .." which looks like a note from original draft that has not been fixed.

Reviewer #2 (Remarks to the Author):

The authors address the reviewer's concern by additional experiments and discussion. The reviewer would recommend the revised form of the text for publication, and encourage the authors to make minor revision of the supporting Information.

S13 How did the authors decide Tg points because the DSC curves did not show clear inflection points. I would see the details in the figure caption.

Reviewer #3 (Remarks to the Author):

The authors adequately amended the manuscript. I recommend publication.

The point by point to the comments of the referees is included below:

Reviewer #1 (Remarks to the Author):

Reading through the revised manuscript I believe that the majority of the referee comments have been addressed satisfactorily. I think the blending aspect of MOF glasses constitutes enough novelty for publication.

The manuscript still contains a few errors that should have been cleaned up by now. For example, in the synthesis part of the methods sections, some volumes are listed with units of Angstroms (should of course be cubic Angstroms) and a few lines later there is a reference to ".....milled for 5 (or 20?) minutes .." which looks like a note from original draft that has not been fixed.

This has been done.

Reviewer #2 (Remarks to the Author):

The authors addresses the reviewer's concern by additional experiments and discussion. The reviewer would recommend the revised form of the text for publication, and encourage the authors to make minor revision of the supporting Information.

S13 How did the authors decide Tg points because the DSC curves did not show clear inflection points. I would see the details in the figure caption.

This has been done, and an extra figure added to further explain this derivation.

Reviewer #3 (Remarks to the Author):

The authors adequately amended the manuscript. I recommend publication.